

**Evaluation of stratospheric age-of-air from CF$_4$, C$_2$F$_6$, C$_3$F$_8$, CHF$_3$, HFC-125, HFC-227ea and SF$_6$; implications for the calculations of halocarbon lifetimes, fractional release factors and ozone depletion potentials.**

**Emma Leedham Elvidge[1], Harald Bönisch[2], Carl A. M. Brenninkmeijer[3], Andreas Engel[4], Paul J. Fraser[5], Eileen Gallacher[1], Ray Langenfelds[5], Jens Mühle[6], David E. Oram[1], Eric A. Ray[7,8], Anna R. Ridley[1], Thomas Röckmann[9], William T. Sturges[1], Ray F. Weiss[6], and Johannes C. Laube[1]**

[1] School of Environmental Sciences, University of East Anglia, Norwich Research Park, Norwich, NR4 7TJ, UK

[2] Institute of Meteorology and Climate Research, Karlsruhe Institute of Technology, Karlsruhe, Germany

[3] Max Planck Institute for Chemistry, Mainz, Germany

[4] Institute for Atmospheric and Environmental Sciences, Goethe University of Frankfurt, Frankfurt, Germany

[5] Climate Science Centre, CSIRO Oceans and Atmosphere, Aspendale, Victoria, Australia

[6] Scripps Institution of Oceanography, University of California, San Diego, La Jolla, California, USA

[7] Chemical Sciences Division, Earth Systems Research Laboratory, NOAA, Boulder, Colorado, USA

[8] Cooperative Institute for Research in Environmental Sciences, University of Colorado, Boulder, Colorado, USA

[9] Institute for Marine and Atmospheric Research Utrecht, Utrecht University, Utrecht, The Netherlands

Contact: e.leedham-elvidge@uea.ac.uk



**Abstract**

In a changing climate, potential stratospheric circulation changes require long-term monitoring. Stratospheric trace gas measurements are often used as a proxy for stratospheric circulation changes via the 'mean age of air' values derived from them. In this study, we investigated five potential age of air tracers – the perfluorocarbons $CF_4$, $C_2F_6$ and $C_3F_8$ and the hydrofluorocarbons $CHF_3$ (HFC-23) and HFC-125 – and compare them to the traditional tracer $SF_6$ and a (relatively) shorter-lived species, HFC-227ea. A detailed uncertainty analysis was performed on mean ages derived from these 'new' tracers to allow us to confidently compare their efficacy as age tracers to the existing tracer, $SF_6$. Our results showed that uncertainties associated with the mean age derived from these new age tracers are similar to those derived from $SF_6$, suggesting these alternative compounds are suitable, in this respect, for use as age tracers. Independent verification of the suitability of these age tracers is provided by a comparison between samples analysed at the University of East Anglia and the Scripps Institution of Oceanography. All five tracers give younger mean ages than $SF_6$, a discrepancy that increases with increasing mean age. Our findings qualitatively support recent work that suggests the stratospheric lifetime of $SF_6$ is significantly less than the previous estimate of 3200 years. The impact of these younger mean ages on three policy-relevant parameters – stratospheric lifetimes, Fractional Release Factors (FRFs), and Ozone Depletion Potentials – is investigated in combination with a recently improved methodology to calculate FRFs. Updates to previous estimations for these parameters are provided.



## 1. Introduction

The 'mean age of air' (mean AoA), defined as the average time that an air parcel has spent in the stratosphere, is an important derived quantity used in several stratospheric research fields, often where direct physical or chemical measurements are scarce, not available or inadequate. AoA is perhaps best known for being a proxy for the rate of the stratospheric mean meridional circulation, the Brewer-Dobson circulation (BDC), as well as being used to determine air mass fluxes between the troposphere and stratosphere (Bönisch et al., 2009). It is also used in calculations to determine the state of recovery of the ozone layer via its role in calculations of stratospheric lifetimes, Ozone Depletion Potentials (ODPs) (Brown et al., 2013; Laube et al., 2013; Volk et al., 1997) and Effective Equivalent Stratospheric Chlorine (Newman et al., 2006).

Mean ages can be derived by comparing an observed abundance of a stratospheric tracer to the tropospheric time series of that gas, assuming that the trace gas in question is largely chemically inert in the stratosphere and has a monotonically, ideally linearly, changing tropospheric concentration (Hall and Plumb, 1994). Commonly used tracers include sulphur hexafluoride ($SF_6$) and carbon dioxide ($CO_2$), which have been used extensively to track large-scale stratospheric transport and transport trends and to evaluate atmospheric residence times of ozone-depleting substances (ODSs) and their impact on the ozone layer (Andrews et al., 2001; Engel et al., 2002; Volk et al., 1997). There are, however, problems with using these compounds as age tracers. The limitations of $CO_2$ have been recently outlined in detail by Engel et al. (2017) and include a complicated tropospheric trend – in part due to the influence of its seasonal cycle (Bönisch et al., 2009) – and a stratospheric $CO_2$ source, i.e. the oxidation of hydrocarbons. For $SF_6$, recent research suggests its lifetime has likely been overestimated, thus it may be giving high-biased mean ages. The reduction in $SF_6$ lifetime comes from both modelling and measurement studies, which have evaluated its stratospheric loss mechanisms via electron attachment (Kovács et al., 2017) and in the polar vortex (Andrews et al., 2001; Ray et al., 2017). The most recent (at time of writing) evaluation gives a revised lifetime of 850 (580-1400) years (Ray et al., 2017). This is considerably lower than the 3200 year lifetime used in the most recent assessments of the Intergovernmental Panel on Climate Change (IPCC) and the World Meteorological Organization (WMO) (IPCC, 2013; WMO, 2014). A revised lifetime will impact the estimated global warming potential of $SF_6$ (Kovács et al., 2017). These limitations do not preclude the use of $CO_2$ and $SF_6$ as age tracers, but may require complex corrections or limit the suitability of these gases to act as tracer in certain regions (Andrews et al., 2001; Bönisch et al., 2009). With this study we do not attempt to discredit these extremely useful existing age tracers, but to add to the range of available tracers to improve the overall understanding in this field.

As mentioned above, AoA is an in important component in our understanding of the BDC. The potential changes to the BDC as the troposphere warms are not yet fully understood. Chemistry-climate models predict an increase in the strength of the BDC (e.g. Li et al., 2008; Oberländer et al., 2013), which would be observed as a negative trend in (or a move to younger) mean ages. However, a time series of mean ages derived from stratospheric observations of trace gases in the mid-latitudes above 25 km has not found a significant trend over the past 40 years (Engel et al., 2009, 2017). Stratospheric circulation is complex: the shallow and deep branches of the BDC may be changing at different rates (Bönisch et al., 2011; Diallo et al., 2012; Ray et al., 2014) and shorter-timescale dynamical changes driven by the Quasi-Biennial Oscillation or the El Niño–Southern Oscillation may complicate or even mask long-term changes to the BDC (Mahieu et al., 2014; Stiller et al., 2017). If a chemical tracer is to be used to diagnose global changes to the BDC it must, therefore, be reliable (that is meeting the criteria of Hall and Plumb (1994), above) throughout the stratosphere. Unfortunately, the influence of $SF_6$-depleted mesospheric air in the upper stratosphere (potential temperature >800 K) and the higher Southern Hemisphere latitudes (poleward of 40 °S) may bias $SF_6$-derived mean ages in these regions (Stiller et al., 2017).

The combination of both the need for accurate age tracers to track stratospheric circulation changes and the uncertainties surrounding existing age tracers prompted us to investigate a suite of anthropogenic trace gases with stratospheric lifetimes >100 years to identify other potential AoA tracers. Of particular interest are the alkane-derived perfluorocarbons (PFCs) which are extremely long-lived, stable trace gases (WMO, 2014), at least one of which, perfluoromethane ($CF_4$), was previously shown to have potential as an age tracer (Harnisch et al., 1999). In this paper, we assess the use of six alternative stratospheric age tracers[1]: $CF_4$, perfluoroethane ($C_2F_6$), perfluoropropane ($C_3F_8$), trifluoromethane ($CHF_3$), pentafluoroethane (HFC-125) and 1,1,1,2,3,3,3-heptafluoropropane (HFC-227ea) and compare them with the existing age tracer $SF_6$. An overview of all compounds discussed in this manuscript, including current stratospheric lifetime estimates and tropospheric growth rates, can be found in Table 1.

---

[1] To enhance the readability of this manuscript we have selected the most common name for each compound to use as its abbreviation, even if this means mixing chemical conventions (e.g. $CHF_3$ but HFC-227ea). Full details for each compound are provided in Table 1.



As well as the potential for expanding the number of chemical species used as stratospheric age tracers the methods available for collecting stratospheric air samples are also increasing. Recently air from the novel AirCore method has been used to calculate $CO_2$-derived mean ages (Engel et al., 2017) and lightweight stratospheric bag samplers have also been developed (Hooghiem et al., 2017). These technologies provide an excellent opportunity to increase the temporal and spatial coverage of stratospheric measurements in an affordable manner. However, it is important that

the mean ages derived from these air samples (which may, in the case of discrete air samples, be as little as 20 ml of air per sample) have a similar level of uncertainty as more traditional samplers (i.e. large balloon-borne cryosamplers and high altitude research aircraft, Sect. 2), especially if we wish to compare changes in mean ages over time. In Sect. 3 we provide details of our uncertainty analysis to facilitate similar analyses on future mean age calculations.

We investigated this set of tracers for a variety of reasons. Firstly, we selected several tracers – $CF_4$, $C_2F_6$, $C_3F_8$ and $CHF_3$ – with estimated stratospheric lifetimes greater than $SF_6$ (Table 1), because of their potential to be suitably-inert age tracers. Secondly, we selected a tracer – HFC-227ea – with a lifetime shorter than (the current established) $SF_6$ lifetime to provide a contrasting point of comparison. Recently, the $SF_6$ lifetime has been shown to be perhaps closer to HFC-227ea than previously thought (Ray et al., 2017, Table 1) and so we include it in our comparison. Finally, we

included HFC-125 as a potential age tracer as we believe its current estimated stratospheric lifetime of 351 years (SPARC, 2013) is potentially an underestimate. We believe the lifetime of HFC-125 (C2, $CHF_2CF_3$) should fall between $CHF_3$ (C1) and HFC-227ea (C3, $CHF_2CF_2CF_3$). All seven of the above-mentioned tracers currently fulfil the prerequisite of having well-constrained monotonically increasing growth rates in the troposphere.

**2. Methodology**
Long-term tropospheric time series are required to define the input of each tracer to the stratosphere. No definition of 'long-term' has been set, but several studies use a period of 10-15 years leading up to the stratospheric measurement period as a suitable tropospheric time series input for mean age calculations of 0-8 years, or even up to 10 years if a time series at the later end of this range is used (Engel et al., 2002, 2006; Stiller et al., 2008). The University of East

Anglia (UEA) has analysed whole air samples from the Cape Grim Baseline Air Pollution Station in Tasmania, Australia (https://agage.mit.edu/stations/cape-grim), since 1978, for all compounds except $CF_4$. The Cape Grim (CG) air archive contains trace gas records known to be representative of unpolluted Southern Hemispheric air and so provides excellent records of globally-relevant tropospheric growth rates (Oram et al., 2012, and references within). UEA trace gas analysis of the CG air archive has been well documented in previous publications, (e.g. Fraser et al.,

1999; Laube et al., 2013)**.** Briefly, analysis is performed using an in-house built manual cryogenic extraction and pre-concentration system connected to an Agilent 6890 gas chromatograph and a high-sensitivity tri-sector mass spectrometer. Full details of the analytical system can be found in Laube et al. (2010a, 2016). Of note is the instrument change detailed in Laube et al. (2016) whereby $C_2F_6$ precision is improved by analysing samples on a KCl-passivated Al-PLOT column, alongside measurements of $SF_6$, $C_3F_8$, $CHF_3$, HFC-125, and HFC-227ea with an Agilent GS

GasPro column. Prior to 2006, analysis was performed on a previous version of the analytical system (still using a GasPro column) that also used different air standards. Data analysed on this older instrument were incorporated into the time series using standard intercomparisons and standard-to-sample ratio comparisons and showed no significant differences. The ions used to quantify the gases measured at UEA were $C_2F_5^+$ (m/z 118.99) for $C_2F_6$, $SF_5^+$ (m/z 126.96) for $SF_6$, $C_3F_7^+$ (m/z 168.99) for $C_3F_8$, $CHF_2^+$ (m/z 51.00) for $CHF_3$, $C_2HF_5+$ (m/z 101.00) for HFC-125 and

$C_3HF_7^+$ (m/z 151.00) for HFC-227ea.

These measurements have been published either as time series or as comparisons to other long-term data sets for $SF_6$ (Laube et al., 2013), $C_2F_6$ (Trudinger et al., 2016), $C_3F_8$ (Trudinger et al., 2016; Ray et al., 2017), $CHF_3$ (Oram et al., 1998), and HFC-227ea (Laube et al. 2010a; Ray et al., 2017). UEA HFC-125 has not been published previously, but

the UEA data agrees very well with Advanced Global Atmospheric Gases Experiment (AGAGE) CG observations (data not shown). Data from high frequency in-situ and archived CG air samples measured by the Scripps Institution of Oceanography (SIO) and the AGAGE network has also been provided for $CF_4$, $C_2F_6$ and $SF_6$. These samples were analysed on a 'Medusa' gas-chromatographic system with cryogenic pre-concentration and mass spectrometric detection (Arnold et al., 2012; Miller et al., 2008). SIO CG $CF_4$ and $C_2F_6$ time series have previously been published in

Mühle et al. (2010) and Trudinger et al. (2016) and their $SF_6$ time series in Rigby et al. (2010). SIO $CF_4$ and $SF_6$ data are reported on the SIO-05 scale and $C_2F_6$ on the SIO-07 scale (Mühle et al., 2010; Prinn et al., 2000).



To ensure suitability of the CG measurements as a record of stratospheric inputs we first compensated for the time lag between observed concentrations in the Southern Hemisphere and the tropical upper troposphere – the main stratospheric input region – by applying a six-month time shift to all CG records. Efficacy of this treatment was verified by comparing the offset CG trends to tropical (20 °N to 20 °S) mid to upper tropospheric aircraft data obtained from interhemispheric flights by the CARIBIC[2] observatory (Fig. 1). As can be seen in Fig. 1, there are some gaps in the UEA CG time series. To smooth the temporal distribution a polynomial fit was applied to each dataset and the equation from this fifth ($CHF_3$, HFC-125, HFC-227ea) or sixth ($SF_6$, $C_2F_6$ or $C_3F_8$) order polynomial fit was used to interpolate monthly mixing ratio values. The fit was applied to the central section of each time series only (see Fig. 1), avoiding periods with significantly different growth rates, e.g. no significant growth for HFC-125 until the mid-1990s. This central section still covered between 81-92% of the UEA CG record for all compounds except $CHF_3$ (58%) and HFC-125 (43%) and provided a suitably-long time series leading up to the stratospheric campaigns (black vertical lines in Fig. 1) for AoA calculations. We were left with a time series between 13-21 years, compound dependent, compared to the 10-15 year time periods utilised in some previous studies (Engel et al., 2002, 2006; Stiller et al., 2008). A bootstrap procedure, outlined below, was used to determine whether polynomial fits were robust throughout the time-period of interest. Two other fit procedures were compared to the polynomials using IGOR Pro software. The cubic spline interpolation failed to cope with the temporally patchy nature of the UEA CG time series and the smoothing spline interpolation provided similar results to the polynomial fits, without the ability to incorporate them into the bootstrap procedure required for our uncertainty analysis. The mean ages derived from the fit-interpolated data were also compared to those derived from the 'raw' CG time series, as used in Laube et al. (2013). The difference between the mean ages derived from these two methods was, for all compounds except HFC-227ea, a maximum of around 2 months (Supplementary Information 2, S2), but the uncertainties associated with the fit-derived mean ages was smaller than those derived from the 'raw' CG dataset (S2). As the SIO CG records had a higher sampling frequency during the period of interest their raw time series were used as inputs into the AoA routine.

Stratospheric measurements in this manuscript were obtained from balloon and aircraft-based whole air-sampling campaigns that took place between 1999 and 2016 (Table 2). The campaigns covered the polar (B34, K2010 and K2011), mid-latitude (OB09, SC16) and tropical (B44) stratosphere. For B44, OB09, K2010, K2011 and SC16 all compounds except $CF_4$ were analysed at UEA on the same system used to analyse the tropospheric trends with B34 $C_2F_6$ samples being analysed on the older version of this instrument. B34 $SF_6$ data were provided by the Goethe University Frankfurt. Sample collection and campaign details for OB09, K2010 and K2011 are discussed in Laube et al. (2013) and OB09 and B44 are discussed in Laube et al. (2010a). The B34 campaign used the same equipment outlined in B44. For more information on the recent StratoClim campaign (SC16) visit http://www.stratoclim.org.

A subset of K2010 and K2011 samples were also analysed at SIO using the Medusa system and calibration scales described above for the AGAGE SIO CG records. SIO provided data for $CF_4$, $C_2F_6$ and $SF_6$. Due to the low pressure and volumes of these samples, only around 280 ml of sample were measured, alternated by the same volume of reference gas. The K2010 samples were at a pressure that allowed for analysis via the standard Medusa method (see references above) using Veriflow clean pressure regulators to sample 6-12 repeated measurements at roughly constant pressures. Due to the lower pressure in the K2011 samples these analysed against an identically-constructed sample flask containing a reference gas at the same pressure as the starting pressure in each K2011 sample. This allowed for both sample and reference gas to be analysed without a regulator and allowed for concurrent pressure decreases in sample and calibration flask, mitigating the possible impact that a difference in pressure between ambient and calibration samples may have had on the SIO analysis. Between 3-8 repetitions were conducted for the K2011 samples. Analytical precisions for SIO data are provided in Table 2.

Uncertainties provided for all UEA measurements are a combination of the analytical precision calculated from repeat analyses of the calibration standard across each analysis day and the regular (usually daily) paired or triplicate analysis of individual samples. Samples where the total uncertainty was greater than three times the standard deviation of the uncertainties across the entire campaign analysis period were excluded. The percentages of samples removed across all campaigns were: ~4% for $SF_6$, $CHF_3$ and HFC-227ea, ~3% for HFC-125, 2% for $C_3F_8$ and none for $C_2F_6$. Datasets provided by other institutions (University of Frankfurt B34 $SF_6$ and SIO K2010 and K2011 data) were smaller and could therefore not be quality controlled in this manner; all data provided to us were included in further analyses.

165

---

[2] CARIBIC (Civil Aircraft for the Regular Investigation of the atmosphere Based on an Instrument Container), part of IAGOS (www.iagos.org) is an observatory based on approximately monthly flights on board a commercial Lufthansa Airbus A340-600 from Frankfurt to destinations on several continents. Further details can be found at http://www.caribic-atmospheric.com/





A sample of stratospheric air represents a mixture of air masses with different transport histories and thus different ages. This distribution of transport times is the 'age spectrum', a probability density function for which the first moment, or mean, is the mean age for that parcel and the second moment, or variance, is the width of the age spectrum (Hall and Plumb, 1994). Mean ages were calculated using the parameterisation described in Bönisch et al. (2009)**.** As described in Engel et al. (2002), stratospheric mixing ratios cannot simply be calculated by propagating the tropospheric trend into the stratosphere: due to nonlinearities in the tropospheric trends for our compounds of interest, the width of the age spectrum impacts the propagation of tropospheric trends into the stratosphere. The width of the age spectrum cannot be measured directly and we assume a constant value of 0.7 as the parameterisation of the ratio $\frac{width\ age\ spectrum^2}{mean\ age}$ (from Hall and Plumb, 1994). This assumption was used in previous studies (Engel et al., 2002; Laube et al., 2013) but to provide a measure of the impact this assumption may have we also compared mean ages calculated using values of 0.5 and 1 (discussed further in Sect. 3d).

**3. Description of and results from the age tracer uncertainty assessment**
As this study focuses on assessing potential new age tracers we carefully consider the uncertainties associated with the mean ages calculated by our AoA routine. Potential sources of uncertainty include: (a) uncertainties in the tropospheric trend; (b) uncertainties in the stratospheric measurements; (c) different methods of implementing the tropospheric trend within the AoA routine; (d) different methods for the parameterisation of the width of the age spectrum. These four main areas of uncertainty are discussed below. A wider suite of tests was performed to help us better understand the mean age uncertainty, many of which have informed our protocol for investigating the main uncertainties components (a-d) or are referenced in our analysis of these components in the following text. Supplementary Information 2 includes a table which provides an overview of the full suite of uncertainty tests performed on our dataset.

For each uncertainty analysis a similar procedure was followed. Here the procedure is outlined using generic terminology, with a specific example in italics.
1. A component of the mean age calculation was identified and considered as the base scenario.
   *We used our Cape Grim raw time series ('raw', the grey markers in Fig. 1) as the tropospheric trend input.*
2. The errors associated with this component were identified.
   *The analytical uncertainty on each of the measurements in the 'raw' time series.*
3. A 'min' and a 'max' dataset was created using these uncertainties.
   *Our mean mixing ratio minus the respective analytical uncertainty value provides the 'raw_min' dataset. Addition of the analytical uncertainty provides 'raw_max'.*
4. A mean age is calculated for each of our stratospheric air samples using the base scenario.
   *Mean ages calculated using 'raw' as the tropospheric input.*
5. Keeping everything else constant (S2) the mean age was calculated again using the 'min' and 'max' datasets.
   *Mean ages calculated using 'raw_min' and 'raw_max' as tropospheric inputs.*
6. The mean ages obtained from 'min' and 'max' are compared to those from the base scenario. Often the difference between the 'min' and 'max' cases are plotted as a 'residual plot'. The average difference between 'min' and 'max' cases is provided in Table 3 (if one of the key uncertainties) or S2 (all tests).
   *The mean ages derived for each stratospheric measurement using 'raw', 'raw_min' and 'raw_max' are compared. The absolute average difference between 'raw' and its min/max variants was 0.5 months for $SF_6$ (case 2 in S2).*

**3a. Uncertainties in the tropospheric measurements**
The first class of uncertainties we consider are those associated with the fit-interpolated tropospheric trend (cases 4 and 5 in SI2). Here our base scenario comprised mean ages derived from the fit-interpolated tropospheric trend (hereafter referred to as 'fit'), compared to those derived from 'fit_min' and 'fit_max', which we obtained from a bootstrap procedure (Efron, 1979; Singh and Xie, 2008). No sampling perfectly represents natural variability and the resampling procedure used during the bootstrapping is designed to provide an indication of the impact of this 'subsampling effect'. Our bootstrap procedure was performed as follows:
1. To enhance our representation of atmospheric variability, we first took our CG time series (Table 1) and converted it to a 3n dataset comprised of [original_data] + [original_data_minus_analytical_uncertainty] + [original_data_plus_analytical_uncertainty]. However, we only resampled a dataset of the original size.
2. We used the bootstap macro for Microsoft Excel provided by Barreto and Howland (2006) to resample (with replacement) our CG dataset. A polynomial fit was applied to each resample.
3. After 1000 iterations, the standard deviation on the fit parameters was calculated.
4. The standard deviation from the bootstrapping procedure was used to create 'fit_min' and 'fit_max' datasets which could be used as tropospheric inputs to the AoA routine.



The ±1 standard deviation uncertainties from this procedure are plotted as dark blue lines in Fig. 1. The uncertainties
associated with the fits are small and show that the polynomials are robust throughout the section of the trend used as
an input into the AoA routine. The mean ages resulting from 'fit_min' and 'fit_max' were compared to the original
mean age values to give an uncertainty estimate for the tropospheric trend components of the AoA routine (Table 3).
Average uncertainties were around 1-3 months. There are some higher values for $C_3F_8$ and HFC-227ea due to the
poorer data coverage in the late 2000s causing the fit to be slightly less robust. This highlights the importance of
ongoing, reliable and regular tropospheric time series measurements for potential new age tracers. These uncertainties
will be combined into an overall uncertainty for each species later in the manuscript.

**3b. Uncertainties in the stratospheric measurements**

As with the tropospheric trends, 'stratmin' and 'stratmax' datasets based on our measurements ± the analytical
uncertainties were used as inputs into the AoA routine and the outputs compared to mean ages derived from the
original stratospheric mixing ratios (cases 8 and 9 in SI2). Results from this comparison are shown as a residual plot in
Fig. 2, where the residuals are the differences between the mean age calculated using our original stratospheric mixing
ratios and those from 'stratmin' and 'stratmax'. The impact of the stratospheric measurement uncertainty is larger than
for the tropospheric inputs: roughly double for $CF_4$, $C_2F_6$, $CHF_3$, HFC-227ea and $SF_6$ and similar for $C_3F_8$ and HFC-
125, but generally averaged around half a year or less for all compounds (Table 3). Differences between different
compounds can be attributed to a combination of their growth rates and their stratospheric measurement precision
(Table 2). The ratio of the stratospheric measurement precision to the growth rate impacts our mean age resolution:
uncertainties derived from our stratospheric measurement precision will be greater if the growth rate is smaller. The
growth rate of $C_2F_6$ was slowing (Fig. 1) in the period leading up to our 2009-2011 campaigns and this is contributing
to the larger uncertainties associated with $C_2F_6$ compared to other compounds, despite similar analytical precisions
(Table 2). For $C_2F_6$ and $SF_6$ there are both UEA and SIO values (Fig. 2, cases 35 and 36 in S2). The mean ages
derived from stratospheric samples analysed by SIO are independent of the UEA measurements, having been
calculated using AGAGE-based tropospheric trends and uncertainties. There are some higher SIO $C_2F_6$ residual values
linked to the higher analytical uncertainty for the SIO measurements (Table 2). This increased uncertainty is not
unexpected: $C_2F_6$ is the least abundant of the three gases measured by SIO for this study and their analytical system is
designed for air samples an order of magnitude, 2 L versus 280 ml, larger than what is available from stratospheric
samples. $SF_6$ measured at both UEA and SIO showed similar stratospheric uncertainties. Independent verification adds
significant weight to the suitability of these new compounds for use as age tracers. The larger impact of uncertainties
in stratospheric data compared to the tropospheric trend (Table 3) highlights the importance of precise measurements
of these compounds if they are to be suitable age tracers. These stratospheric uncertainties are combined with
uncertainties from Sect. 3a to create an overall uncertainty later in the manuscript.

**3c. Comparing different methods for implementing the tropospheric time series component of the mean age
calculation**

We used an AoA routine based on the algorithm described in Engel et al. (2009), based on the method provided for
inert tracers by Hall and Plumb (1994). The limitation of this method is that only a quadratic function can be applied
for fitting the tropospheric time series for the AoA calculation. A recent improvement is to calculate AoA by a
numerical method that uses the convolution of the age spectra, approximated by an inverse Gaussian distribution with
the tropospheric time series (Ray et al., 2017), which overcomes the limitations of a quadratic fit to approximate such
trends. We implemented this numerical convolution method in our AoA routine so that we could compare mean ages
derived from our data using both the original quadratic and the numerical convolution algorithms (cases 18 in S1). The
resulting 'residual plot' can be seen in Supplementary Information 3 (S3) and the average uncertainties in Table 3. We
found that outside of very young (<1 year) mean ages the difference between these two methods was one month or
less. The weaker performance near the tropopause is a known problem of the convolution method for younger mean
ages, which require the convolution over a short time period, potentially leading to mean age biases due to observed
short-term variability and/or data sparsity. As the quadratic method performed better across the whole range of mean
ages in our study, we use that method to derive mean ages and uncertainties discussed in all subsequent sections of the
manuscript.



### 3d. Uncertainty in parameterisation of width of age spectrum

As described in Sect. 2, we used a value of 0.7 as the parameterisation of the ratio between the squared width of the age spectrum and the mean age to assist with the propagation of non-linear tropospheric trends into the stratosphere.

Previous studies have investigated the effect of varying this parameterisation. Engel et al. (2002) investigated the impact of using values of 0, 0.7 and 1.25 and found differences of less than half a year for $CO_2$ and $SF_6$ mean ages. They also reported that the best agreement between these two age tracers was reached when using 0.7. Laube et al. (2010b) also tested the impact of this value on calculated Fractional Release Factors (FRFs, see Sect. 5), comparing values of 0.5, 0.7 and 1.25 and found this factor had a small impact on the FRF for a range of long-lived halocarbons.

As this study introduces new potential age tracers, investigating the impact of this parameterisation is pertinent. Values of 0.5 and 1 were compared to the commonly-used value of 0.7 (residual plot in S3). The results are shown in Table 3: one can see that the impact is small (< 1 month, on average) compared to the impact of (a) and (b), and is similar for all compounds.

### 4. Combination of errors and analysis of new age tracers

The two key uncertainties from Sect. 3, namely those associated with the tropospheric trend and stratospheric measurements (columns a and b in Table 3), were combined and used as the error bars in Fig. 3, which shows a vertical profile of the mean ages derived from all six of our tracers. We use CFC-11 as a vertical coordinate because it is an inherent property of the measured air parcel and will be similarly influenced by localised transport and mixing.

Tropospheric CFC-11 mixing ratios have slowly declined in the period covered by the stratospheric campaigns (1999-2011) at a rate of between 0.5-1% per year (based on our CG trend). A linear fit of the data throughout this period was relatively robust: ~3% standard deviation between fits calculated over eight different time windows and $R^2$ values of >0.99 for all eight fits. Based on this we corrected the CFC-11 mixing ratios for the stratospheric campaigns relative to the earliest (B34 in 1999) campaign. This is a simplification, as the propagation of tropospheric mixing ratios into

stratosphere is influenced by the width of the age spectrum (see Sect. 2). As the CFC-11 mixing ratios are not used in further calculations (purely as a visual indicator of altitude) and the trend during the time period covered is linear and small, we felt it a suitable approximation for our needs.

As mentioned before, a suitable age tracer must have a well-quantified, monotonically changing tropospheric trend,
precise stratospheric measurements and be relatively inert in the stratosphere. The suitability of our new age tracers to meet the first two requirements is shown by the error bars in Fig. 3 and the final column in Table 3. The uncertainties of the new age tracers were compared to those associated with $SF_6$ and were found to be similar for $C_3H_8$ and HFC-227ea, smaller for HFC-125 and larger, but within a similar magnitude range for $CF_4$, $C_2F_6$ and $CHF_3$. In this respect, these new age tracers are as suitable as the commonly-used tracer $SF_6$. As for the final point, that the compounds are

inert in the stratosphere (suggested by their lifetimes: see Table 1), this is also supported by Fig. 3 where we can compare the mean ages derived from the new tracers to those derived from $SF_6$. It is interesting that $SF_6$ (current lifetime estimate 3200 years) lies to the right of the plot, the trend line in Fig. 3a overlapping with HFC-227ea (stratospheric lifetime estimated at 673 years). This high bias in $SF_6$-derived mean ages supports the recently revised $SF_6$ lifetime estimate of 850 (580-1400) years (Ray et al., 2017). The other compounds tend to give younger mean

ages consistent with longer stratospheric lifetimes. In particular, HFC-125 shows evidence of having a stratospheric lifetime well in excess of 351 years (see Sect. 1). Loss of $SF_6$ may be understandable in the polar regions during winter due to the mesospheric sink and the downward transport of $SF_6$ depleted mesospheric air within the polar vortex, but when we split our results into polar (Fig. 3b) and mid-latitude and tropical (Fig. 3c) flights one can see that the $SF_6$ fit still mimics that of HFC-227ea, suggesting there is evidence, even in this region, that $SF_6$-derived mean

ages may be more consistent with the shorter-lived HFC-227ea. This raises the question as to whether the sink of $SF_6$ is indeed exclusively located in the mesosphere, although admittedly our non-polar dataset is limited and we cannot rule out mixing of polar vortex air (or vortex remnants) being observed in mid-latitudes outside of the winter polar vortex (Strunk et al., 2000).

Table 4 shows the degree of agreement, within stratospheric measurement uncertainties (column b in Table 3), of the mean ages derived from each of the age tracers. There is strong agreement between all the new age tracers: $CF_4$, $C_2F_6$, $C_3H_8$, $CHF_3$ and HFC-125. Mean ages derived from these compounds, except for $CHF_3$, do not agree well with the mean ages derived from $SF_6$ and HFC-227ea. With the lifetime of $CHF_3$ in the middle of our range of tracer lifetimes (Table 1) we would expect $CHF_3$-derivded mean ages to agree with both shorter- and longer-lived compounds. There

is good agreement between HFC-227ea and $SF_6$. Table 4 also shows the degree of agreement when the data are split into polar and mid-latitude and tropical datasets. There are less data for the latter group where we have co-measurements of two or more age tracers. However, there is still good evidence that the agreement between $SF_6$ and HFC-227ea is stronger than for $SF_6$ and the new age tracers.





We combined the results from the new age tracers ($CF_4$, $C_2F_6$, $C_3F_8$, $CHF_3$ and HFC-125) to derive a new 'best estimate' of the mean age of air and plotted this against the $SF_6$ mean age in Fig. 4. As we may expect different results in the tropics, the input region to the stratosphere, we have removed our four tropical measurements from our dataset and this slightly reduced dataset is listed as 'all (no tropical)' hereafter. A bivariate linear regression is included for the whole (no tropical) dataset. Bivariate regression fits using only polar, mid-latitudinal, or tropical data (also in Fig. 4)

do not result in significantly different slopes (although the tropical fit exhibits large uncertainties as it is based on four points only). Both Figs. 3 and 4 show that the agreement between $SF_6$ and the other tracers weakens for older mean ages. This is similar to the relationship between mean ages derived from $CO_2$ and $SF_6$ which has been shown to be "excellent" for mean ages up to 3 years by Andrews et al. (2001) and to agree within errors, that is within <0.6 years difference, with Engel et al. (2002). Interestingly, although we do not have $CO_2$ data for our campaigns, the slope in

Fig 4 is remarkably similar to the ~0.8:1 slope derived by Andrews et al. (2001), who compared mean ages of air derived via $SF_6$ and $CO_2$. Within our 'all (no tropical)' dataset our 'best estimate' mean age agreed, within uncertainties, with $SF_6$-derived mean age 63% of the time for mean ages <4 years, 42% of the time within the Engel et al. (2002) window of 2-5 years and only 16% of the time above 5 years. Our results suggest that care should be taken when using $SF_6$ as an age tracer for older (high altitude) air where its loss processes (Sect. 1) may bias derived mean

ages. The smaller sample size with mean ages less than 3 years (n=33 compared to n=112 over 3 years) makes it difficult to conclude if this bias exists in samples with $SF_6$-derived young mean ages. However, Fig. 4 shows that when the fit is applied only to samples with $SF_6$ mean ages < 3 years it is, for the most part, similar (within uncertainties) to that derived from the complete dataset.

Fig. 4 also includes SC16 data: recently-analysed mid-latitude data from two aircraft flights in the Mediterranean region (Table 2). Stratospheric uncertainties (as outlined in Sect. 3b) were calculated for SC16 samples in the same manner as for other compounds. As our existing selection of high-altitude campaigns only included two mid-latitude and one tropical flight (the latter comprising of only four data points) we thought it important to include these data. However, the SC16 samples are not discussed in the error analysis above for two reasons. Firstly, the target of this

campaign was to sample polluted air from the Asian monsoon outflow. The impact of pollution can be seen in the high levels of several gases, including $SF_6$, near the tropopause (all but three samples were collected at potential temperatures >380 K). Secondly, the estimation of mean ages near the tropopause is limited by the availability of our CG-based tropospheric trend, which currently ends in February 2017. As that trend needs to be shifted by 6 months to account for interhemispheric transport (see Sect. 2) it only just extends to the time of these flights, increasing the

uncertainties associated with the polynomial fits (Sect. 2). As high levels of $SF_6$, or other age tracers, biases the derived mean ages toward younger values, the more uncertain mean ages (<0.5 years) were removed for Fig. 4 and further analysis. Despite these differences, the slope of $SF_6$-based vs 'best estimate'-based mean ages for SC16 is similar to that of the other campaigns.

**5. Implications for policy-relevant parameters**
Younger mean ages do have implications for three important policy-relevant parameters that are used to quantify the impact of halocarbons on stratospheric ozone:
   a. Stratospheric lifetimes of ODSs.
   b. FRFs: the fraction of a halocarbon that has been converted into its reactive (ozone-depleting) form in the
stratosphere. Compounds with larger FRFs result in greater ozone depletion.
   c. ODPs: a measure of the impact of individual halocarbons to deplete ozone relative to CFC-11.

These three parameters were calculated using $SF_6$-based mean ages in Laube et al. (2013). We revisit this dataset here, comparing the Laube et al. results to updated FRFs, lifetimes and ODPs calculated from our new 'best estimate' mean

age derived from our five new age tracers for the following 10 ODSs: CFC-11, CFC-113, CFC-12, HCFC-141b, HCFC-142b, HCFC-22, Halon-1301, Halon-1211, carbon tetrachloride ($CCl_4$) and methyl chloroform ($CH_3CCl_3$).

**5a. Stratospheric lifetimes derived from new age tracers**
The lifetime of the ten ODSs listed above were calculated in Laube et al. (2013) using a method dependent on the
390 slope of the correlation between CFC-11 mixing ratios and mean ages at the tropopause. When using the new 'best estimate' mean age estimate this slope estimate changes from $-20.6 \pm 4.6$ ppt yr$^{-1}$ to $-28.6 \pm 4.3$ ppt yr$^{-1}$. The updated stratospheric lifetimes calculated from our new slope are shown in Table 5, alongside the old values as well as recommendations from WMO (2014). As our lifetime calculation only produces lifetimes relative to that of CFC-11, changes are generally small. The exceptions are the three main hydrochlorofluorocarbons (HCFCs), for which the

395 lifetime has decreased significantly, and $CH_3CCl_3$ for which it has increased. This is linked to the relatively large changes (increases for HCFCs and a decrease for $CH_3CCl_3$) in the tropospheric abundances of these gases in recent years.



### 5b. Fractional Release Factors derived from new age tracers

Two updates to the FRFs reported in Laube et al. (2013) were made and the resulting FRFs can be seen in Table 6, alongside previous results and recommendations from WMO (2014). The first change was to use our new 'best estimate' mean age in the FRF calculation. The second change was to use the new methodology outlined in Ostermöller et al. (2017). Based on the work of Plumb et al. (1999) they presented a new formula to calculate FRFs that considers the dependency of the age spectrum on the stratospheric lifetime and tropospheric trend of the ODS in
question. We applied this correction, using the exact parameterisation suggested by Plumb et al. (1999). We note that some of the lifetimes used by Plumb et al. are somewhat different to ours, but tests on the influence of lifetime on FRFs derived from this parametrisation showed that the impact was limited to ±0.03, which is well within our FRF uncertainties (Table 6). Changes from the initial mean age correction are significant and would result in increased FRFs throughout. However, these two corrections can have contrary effects for species with strongly increasing (e.g.
HCFC-22, Fig. 5b) or decreasing ($CH_3CCl_3$, Fig. 5c) tropospheric abundances. For HCFC-22 the two corrections work in the same direction, resulting in substantially higher FRFs at a given mean age. For $CH_3CCl_3$ the opposite is true and we see very little change.

### 5c. Ozone Depletion Potentials derived from new age tracers

ODPs were calculated relative to CFC-11 using the method in Laube et al. (2013) but with updated tropospheric lifetimes from WMO (2014), the latter mainly affecting compounds with significant removal in the troposphere. As ODPs were calculated relative to CFC-11 (FRF changes shown in Fig. 5a), changes to ODPs are only significant for the three hydrofluorocarbons (HCFCs), which have strong positive trends and thus the largest changes to their FRFs. Our full set of updates can be seen in Table 7. The new HCFC ODP values are now closer to the recommended values
in WMO (2014), and we see agreement between HCFC-141b and HCFC-22 within our uncertainties. Nevertheless, for all other ODSs, except $CH_3CCl_3$, we still find ODPs significantly different to the ones used in WMO (2014). This is even the case when we increase our CFC-11 lifetime to 60.2 years, the equivalent of assuming a CFC-12 lifetime of 102 years as recommended in WMO (2014). However, WMO (2014) values are based on Newman et al. (2006) and do not include the recent correction by Ostermöller et al. (2017). What is also noteworthy from Figure 5 is that the
discrepancy between the FRF-mean age correlations derived by Newman et al. (2006) and Laube et al. (2013) largely disappears with our updates. This confirms the suspicion mentioned in Laube et al. (2013) that this discrepancy might predominantly arise from the use of different age tracers (Newman et al. used $CO_2$-derived mean ages).

### 6. Conclusions

We have presented tropospheric trends and stratospheric measurements of seven trace gases and evaluated their capability to estimate stratospheric mean ages, which are useful proxies for stratospheric transit times. We find that these gases have suitable tropospheric growth rates and measurement precisions (<2% for all compounds across all stratospheric campaigns) for this purpose. A comprehensive uncertainty analysis was performed on several factors contributing to the uncertainties in tracer-derived mean ages. Uncertainties in AoA estimates based on our new tracers
were approximately equal to or less than 6 months for all compounds, similar to those for the existing tracer $SF_6$. In addition, independent analysis of three gases ($CF_4$, $C_2F_6$, and $SF_6$) at SIO using different calibration scales and independent tropospheric trends resulted in very similar mean ages. Importantly, five of these gases, $CF_4$, $C_2F_6$, $C_3F_8$, $CHF_3$, and HFC-125, produce very similar mean ages of air, allowing us to produce a new 'best estimate' mean age which we compared to $SF_6$-derived mean ages. Whilst our non-polar dataset is limited, we provide some qualitative
evidence to suggest potential $SF_6$ loss outside of the polar vortex, and support recent work which suggests a reduction in the $SF_6$ stratospheric lifetime from 3200 to 850 years (Ray et al., 2017). The discrepancy between $SF_6$ and 'best estimate'-derived mean ages is greater for older air, as seen for the $CO_2$-$SF_6$ relationship in Andrews et al. (2001), Engel et al. (2002, 2006) and Ray et al. (2017), although somewhat in disagreement with Strunk et al. (2000) who found that $SF_6$ and $CO_2$ mean ages were consistent up to mean ages of around 7-8 years. Further data from
stratospheric balloon and aircraft flights are needed to answer this question in the future.

The new tracers identified here are not meant to replace $SF_6$ and $CO_2$, which are established age tracers with well-defined tropospheric trends and a wealth of stratospheric measurements, in particular as they are measurable by satellite (Stiller et al., 2008). However, we do provide substantial new evidence for the need of caution when using
$SF_6$ to derive mean ages, especially above the lowermost stratosphere. We also note that, unlike $CO_2$, our new age tracers do not have large seasonal cycles or stratospheric sources and are therefore better suited as tracers of transport times in the lower stratosphere. As future changes to the BDC are likely to be complex, a suite of tracers may be better suited than $SF_6$ or $CO_2$ alone in diagnosing long term changes.





Finally, we use a new tracer-derived 'best estimate' mean age and investigate the knock-on effects on policy-relevant parameters such as stratospheric lifetimes, FRFs and ODPs of 10 important ODSs. A substantial decrease in the lifetime estimates for HCFC-22, -141b and -142b and an increase in that of $CH_3CCl_3$ are observed when compared to the previous $SF_6$-age based estimate of Laube et al. (2013). These changes do not cause large changes to the total
atmospheric lifetimes of these gases, however, as their main sink is the reaction with the OH radical in the troposphere. Our FRF and ODP calculations were further improved by the addition of a recent correction presented in Ostermöller et al. (2017). The interaction between these corrections is complex, but, again only results in substantial, but within ODP calculation uncertainties, changes for the three HCFCs (larger ODPs) and $CH_3CCl_3$ (smaller ODP) compared to Laube et al. (2013). Changes for all four compounds place our ODP estimates closer to the recommended
ODPs in WMO (2014) than the values published in Laube et al., 2013.

**Competing interests**
The authors declare that they have no conflict of interest

**Acknowledgements**
Johannes Laube received funding from the UK Natural Environment Research Council (Research Fellowship NE/I021918/1) and David E. Oram from the National Centre for Atmospheric Science. Part of this work was funded by the ERC project EXC3ITE (EXC3ITE-678904-ERC-2015-STG).We acknowledge the Cape Grim staff over many years for the collection of the Cape Grim Air Archive and for collecting air samples for UEA. Funding for the Cape
Grim Air Archive is from CSIRO, the Bureau of Meteorology and Refrigerant Reclaim Australia. We thank Michel Bolder for collecting the Geophysica air samples and acknowledge the work of the Geophysica aircraft and CNES balloon teams as well as related funding from ESA (PremierEx project), the Forschungszentrum Jülich, the European Commission (FP7 projects RECONCILE-226365-FP7-ENV-2008-1 and StratoClim-603557-FP7-ENV-2013-two-stage), and the Dutch Science foundation (NWO, grant number 865.07.001). The operation of the AGAGE
instruments at SIO is supported by the National Aeronautics and Space Administration (NASA) (grants NAG5-12669, NNX07AE89G, and NNX11AF17G to MIT and grants NNX07AE87G, NNX07AF09G, NNX11AF15G, and NNX11AF16G to SIO).





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



**Figures**

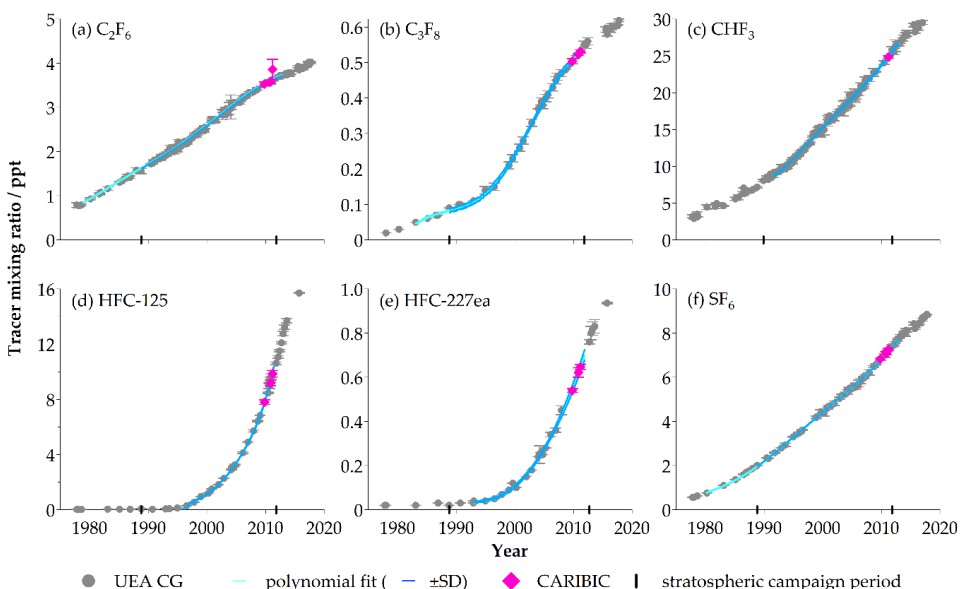

**Figure 1.** UEA CG time series (six month time-shift), polynomial fits applied to these time series, and associated errors (see inset legend). Details of the analytical uncertainties on UEA CG time series, application of polynomial fit and comparison with CARIBIC data are provided in Section 2. Vertical black lines on the x-axis show the section that includes a ten-year period leading up to each of the stratospheric campaigns used during the bootstrap procedure (Section 3a).





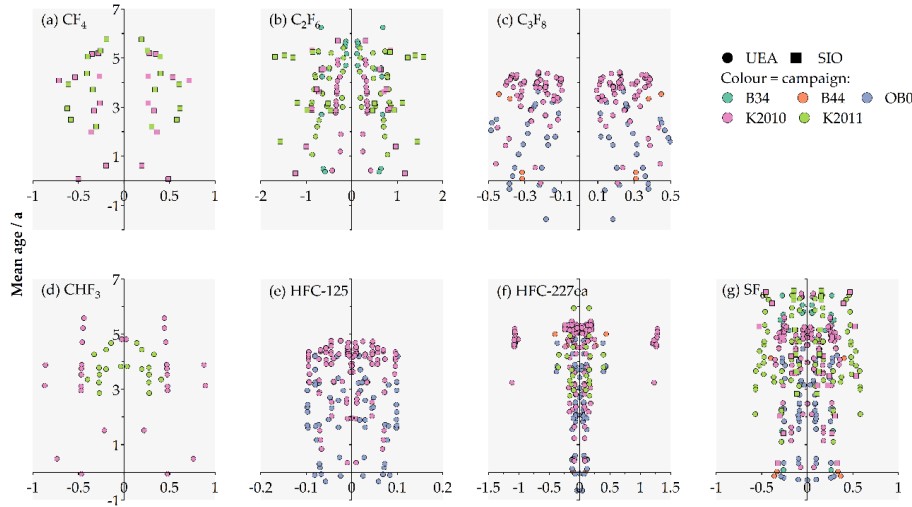

**Figure 2.** 'Residual plots' showing the uncertainties associated with varying the stratospheric measurement inputs for the AoA routine. X-axis shows the difference between the mean ages calculated using a minimum and maximum stratospheric mixing ratio compared to using the mean mixing ratio normally used, the mean age of which is on the y-axis (Section 3b, S2 cases 8 and 9). Marker shape denotes which institution performed the analysis and marker colour the stratospheric campaign, see inset legend. Vertical axis labels for each row are in the left panel.







**Figure 3.** Vertical profiles of mean ages derived from all compounds used in this study. Panels (b) and (c) show the same data as in (a) but split into polar (b) and mid-latitude and tropical (c) flights only (see Table 2 for campaign details). Colours represent different age tracers, see inset legend, and remain the same across all panels.



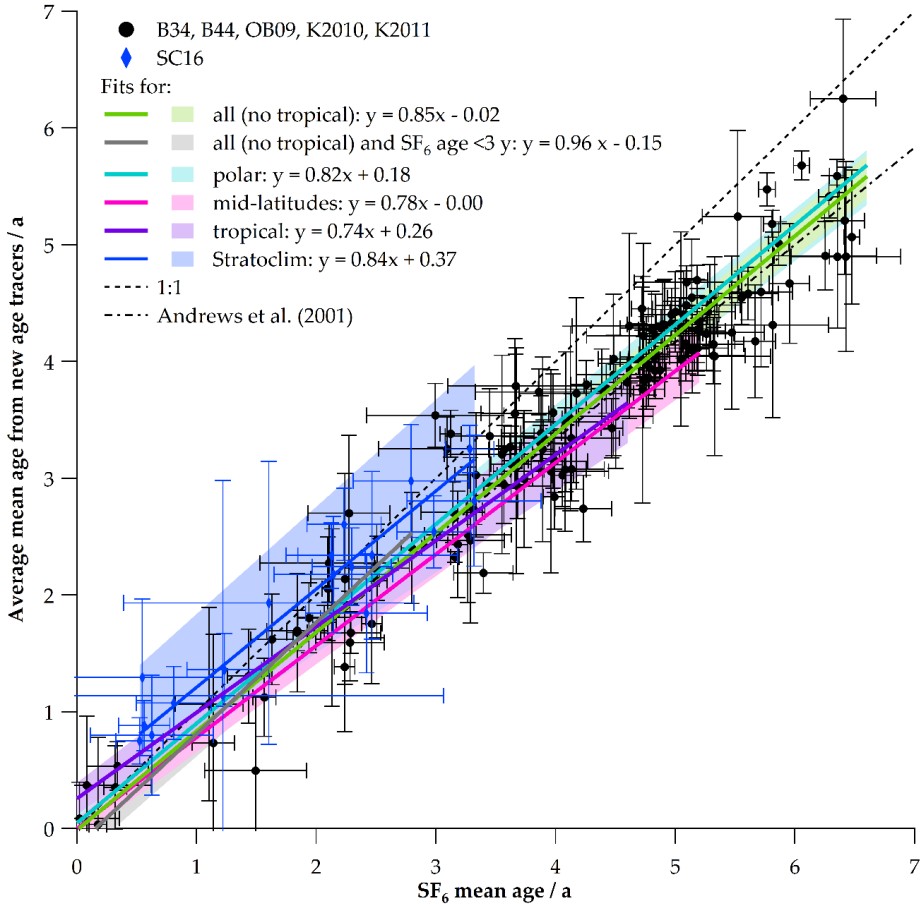

**Figure 4.** 'Best estimate' mean ages (a combined mean age based on $CF_4$, $C_2F_6$, $C_3F_8$, $CHF_3$ and HFC-125) plotted against $SF_6$ mean age. Error bars are based on stratospheric uncertainties from Table 3 column b. All fits are bivariate linear fits with uncertainties shown by shaded areas (see inset legend). $SF_6$ vs $CO_2$ line from Andrews et al. (2001) included for comparison.





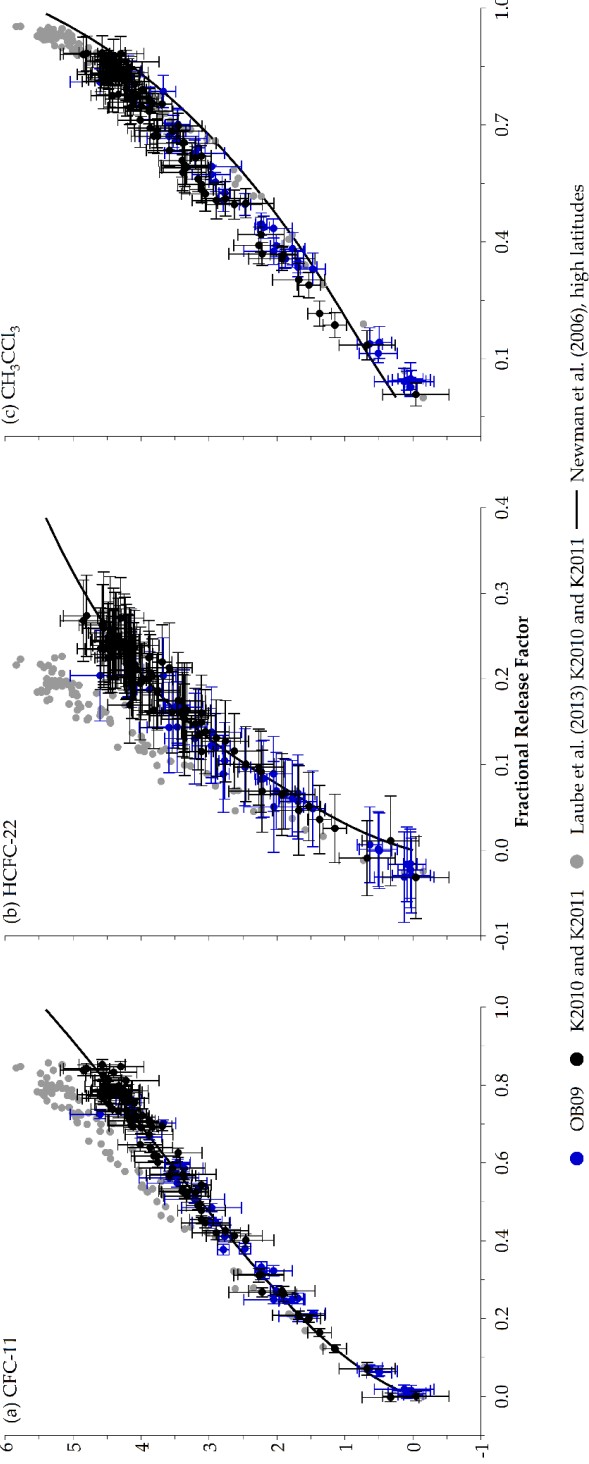

**Figure 5.** Changes in FRFs resulting from our new 'best estimate' mean age of air as well as the improved FRF calculation method from Ostermöller et al. (2017) for OB09, K2010 and K2011, compared to previously published K2010 and K2011 data (Laube et al., 2013) and FRFs-mean age correlations from Newman et al. (2006). Shown for three compound case studies, see details in main manuscript.





**Table 1.** Overview of trace gases used in this study and their relevant properties

| Compound | Formula | Stratospheric lifetime / a (WMO, 2014) | Growth rate / %[a] | Average measurement precision / %[b] | Number of samples in tropospheric time series |
|---|---|---|---|---|---|
| Perfluoromethane, PFC-14 | $CF_4$ | >50,000[c] | 0.90 | 0.2 | 219 |
| Perfluoroethane, PFC-116 | $C_2F_6$ | >10000 | 2.8 | 1.6 | 114 |
| Perfluoropropane, PFC-218 | $C_3F_8$ | ~7000 | 7 | 1.9 | 34 |
| Trifluoromethane, HFC-23 | $CHF_3$ | 4420 | 4.2 | 1.7 | 117 |
| Pentafluoroethane, HFC-125 | $C_2HF_5$ | 351 | 17 | 1.1 | 40 |
| Heptafluoropropane, HFC-227ea | $C_3HF_7$ | 673 | 14 | 2.8 | 29 |
| Sulfur hexafluoride | $SF_6$ | 3200 (850[d]) | 4 | 1.1 | 59 |

[a] Growth rates are annual values averaged from 2002-2012 and derived from our own records, apart from $CF_4$, which is from the SIO AGAGE CG time series 2004-2017 (Section 2), and $SF_6$, where higher frequency 2004-2014 NOAA data are used (see Supplementary Information 1 for agreement between NOAA and UEA data).
[b] Precision calculations are outlined in Section 2. Here the precision is calculated only for the tropospheric time series data. Stratospheric sample precisions are in Table 2.
[c] Total atmospheric lifetime.
[d] Ray et al. (2017).



**Table 2.** Overview of stratospheric campaigns used in this study

| Abbreviation | Campaign dates | Platform | Location Altitude[a], Latitude, longitude Campaigns, collaborations | Data availability — Grey squares = data available. % analytical precision shown where data are available | | | | | | |
|---|---|---|---|---|---|---|---|---|---|---|
| | | | | $CF_4$ | $C_2F_6$ | $C_3F_8$ | $CHF_3$ | HFC-125 | HFC-227ea | $SF_6$ |
| B34 | 06-Feb-99 | High altitude balloon-borne whole air sampler | Kiruna, Sweden Up to 26 km, 62-77 °N, 1 °W-29 °E | | 1.8 | | | | | 2.1 |
| B44 | 11-Jun-08 | | Teresina, Brazil Up to 33.5 km, 5° S, 43° W Launched by the French Space Agency Centre National d'Etudes Spatiales. | | | 2.4 | | | 3.1 | 1.5 |
| OB09 | 30-Oct-09 04-Nov-09 | M55 Geophysica high altitude aircraft | Oberpfaffenhofen, Germany 10-20 km, 48-54 °N, 7-12 °E | | | 0.8 | | 0.3 | 1.6 | 0.5 |
| K2010 UEA | 20-Jan-10 and 02-Feb-10 | | Kiruna, Sweden 9-19 km, 62-77 °N, 1 °W-29 °E | | 0.4 | 1.1 | 1.4 | 0.7 | 1.5 | 0.7 |
| K2010 SIO | | | Part of RECONCILE (von Hobe et al., 2013) and ESSENCE campaigns. | 0.3 | 2.0 | | | | | 1.4 |
| K2011 UEA | 11-Dec-11 and 16-Dec-11 | | | | 1.5 | | 0.6 | | 1.1 | 1.2 |
| K2011 SIO | | | | 0.4 | 2.5 | | | | | 1.3 |
| SC16 | 01-Sep-16 and 06-Sep-16 | | Kalamata, Greece 10-21 km, 33-41 °N, 22-32 °E Part of EU StratoClim project. | | 0.7 | 1.3 | 0.5 | 1.0 | 0.8 | 0.8 |

[a] Maximum sampling altitude for balloons and cruising altitude range for aircraft





**Table 3.** Uncertainties[a] associated with calculating the mean age of air for stratospheric samples

| Compound | (a) Tropospheric trend uncertainties | (b) Stratospheric measurement uncertainties | (c) 'Quadratic' vs 'convolution' AoA routines | (d) Uncertainty in parameterisation of width of age spectrum | Combined uncertainty (a + b only) |
|---|---|---|---|---|---|
| | | ± uncertainties / months mean (min – max) | | | |
| $CF_4$ SIO | 2.1 (1.2–2.5) | 4.7 (2.3–8.6) | - | - | |
| $C_2F_6$ | 1.8 (1.6–2.2) | 5.8 (2.1–10.6) | 0.6 (<0.1–1.0) | 0.7 (0.1–1.2) | 6.0 (2.8–10.6) |
| $C_2F_6$ SIO | 4.2 (3.5–5.1) | 11.1 (3.6–20.2) | - | - | |
| $C_3F_8$ | 2.5 (1.9–4.3) | 3.2 (1.1–6.8) | 1.0 (0.4–1.3) | 0.7 (<0.1–1.0) | 3.7 (2.5–7.2) |
| $CHF_3$ | 1.5 (1.3–1.7) | 4.5 (0.3–10.7) | 0.1 (<0.1–0.2) | 0.3 (<0.1–0.5) | 4.9 (1.4–10.7) |
| HFC-125 | 0.6 (<0.1–0.8) | 0.6 (<0.1–1.2) | 0.6 (<0.1–1.2) | 0.5 (<0.1–1.4) | 0.9 (0.3–1.4) |
| HFC-227ea | 2.4 (1.8–3.2) | 2.9 (0.4–15.4) | 0.2 (<0.1–0.9) | 0.4 (<0.1–1.4) | 4.2 (2.2–14.3) |
| $SF_6$ | 1.1 (0.4–1.9) | 2.5 (<0.1–7.0) | 0.2 (<0.1–0.7) | 0.3 (<0.1–0.5) | 2.8 (1.1–7.0) |
| $SF_6$ SIO | 1.6 (1.3–5.0) | 2.8 (1.3–6.5) | - | - | |

[a]These are averages from campaigns B44, OB09, K2010 and K2011 (Table 2). B34 data are not included as the analysis of these samples was performed on an older instrument ($C_2F_6$) or not at UEA ($SF_6$). SC11 data are not included as a full uncertainty analysis was not performed on SC16 due to the complex air sample source region (Section 4).



**Table 4.** Percentage of samples where the mean age derived from two tracers agreed within the uncertainties[a].

| Shading bands | |
| --- | --- |
| 0-20% | |
| 20-40% | |
| 40-60% | |
| 60-80% | |
| 80-100% | |

**ALL DATA**

**Percentage agreement**

| | CF4 | C2F6 | C3F8 | CHF3 | HFC-125 | HFC-227ea | SF6 |
| --- | --- | --- | --- | --- | --- | --- | --- |
| CF4 | | 93 | | | | 40 | 35 |
| C2F6 | | | | 77 | | | 56 |
| C3F8 | | | | 93 | 76 | 46 | 34 |
| CHF3 | | | | | | 84 | 70 |
| HFC-125 | | | | | | 32 | 15 |
| HFC-227ea | | | | | | | 89 |
| SF6 | | | | | | | |

**Number of samples with measurements of both compounds**

| | CF4 | C2F6 | C3F8 | CHF3 | HFC-125 | HFC-227ea | SF6 |
| --- | --- | --- | --- | --- | --- | --- | --- |
| CF4 | | 15 | 6 | 13 | 5 | 10 | 17 |
| C2F6 | | | 9 | 14 | 8 | 9 | 48 |
| C3F8 | | | | 9 | 91 | 92 | 92 |
| CHF3 | | | | | 8 | 19 | 23 |
| HFC-125 | | | | | | 87 | 88 |
| HFC-227ea | | | | | | | 98 |
| SF6 | | | | | | | |

**MID-LATITUDE AND TROPICAL DATA**

**Percentage agreement**

| | CF4 | C2F6 | C3F8 | CHF3 | HFC-125 | HFC-227ea | SF6 |
| --- | --- | --- | --- | --- | --- | --- | --- |
| CF4 | | | | | | | |
| C2F6 | | | | | | | |
| C3F8 | | | | | 76 | 50 | 46 |
| CHF3 | | | | | | | |
| HFC-125 | | | | | | 47 | 27 |
| HFC-227ea | | | | | | | 82 |
| SF6 | | | | | | | |

**Number of samples with measurements of both compounds**

| | CF4 | C2F6 | C3F8 | CHF3 | HFC-125 | HFC-227ea | SF6 |
| --- | --- | --- | --- | --- | --- | --- | --- |
| CF4 | | 0 | 0 | 0 | 0 | 0 | 0 |
| C2F6 | | | 0 | 0 | 0 | 0 | 0 |
| C3F8 | | | | 0 | 33 | 34 | 37 |
| CHF3 | | | | | 0 | 0 | 0 |
| HFC-125 | | | | | | 30 | 33 |
| HFC-227ea | | | | | | | 33 |
| SF6 | | | | | | | |

**POLAR DATA**

**Percentage agreement**

| | CF4 | C2F6 | C3F8 | CHF3 | HFC-125 | HFC-227ea | SF6 |
| --- | --- | --- | --- | --- | --- | --- | --- |
| CF4 | | 93 | | | | 40 | 35 |
| C2F6 | | | | 77 | | 56 | 56 |
| C3F8 | | | | 93 | 76 | 43 | 25 |
| CHF3 | | | | | | 84 | 70 |
| HFC-125 | | | | | | 26 | 7 |
| HFC-227ea | | | | | | | 92 |
| SF6 | | | | | | | |

**Number of samples with measurements of both compounds**

| | CF4 | C2F6 | C3F8 | CHF3 | HFC-125 | HFC-227ea | SF6 |
| --- | --- | --- | --- | --- | --- | --- | --- |
| CF4 | | 15 | 6 | 13 | 5 | 10 | 17 |
| C2F6 | | | 9 | 14 | 8 | 9 | 48 |
| C3F8 | | | | 9 | 58 | 58 | 55 |
| CHF3 | | | | | 8 | 19 | 23 |
| HFC-125 | | | | | | 57 | 55 |
| HFC-227ea | | | | | | | 65 |
| SF6 | | | | | | | |



[a] Number of data points compared are in the right-hand panel. Percentages are not included where there are less than 10 paired data points available for comparison. The same number of pairs are not available for each compound as not every compound was measured during each campaign (Table 2) and even within a campaign different analytical requirements for different compounds meant not all compounds where reported for each sample (Section 2 and refs within).



**Table 5.** Updated stratospheric lifetimes based on 'best estimate' mean ages derived in this study compared to existing literature values

| Compound | Stratospheric lifetime[a] / a (min-max) | | |
| :---: | :---: | :---: | :---: |
| | **This study** | **Laube et al. (2013)** | **WMO (2014)** |
| CFC-11 | 60 (54-67)[$] | 60 (54-67)[$] | – |
| CFC-113 | 83 (75–94) | 82 (74–93) | 88.4 |
| CFC-12 | *(102)* | *(100)* | 95.5 |
| HCFC-141b | 101 (64–221) | 122 (70–454) | 72.3 |
| HCFC-142b | 178 (103–459) | 406 (139–∞) | 212 |
| HCFC-22 | 129 (94–204) | 184 (113–647) | 161 |
| Halon-1301 | 78 (72–85) | 82 (75–93) | 73.5 |
| Halon-1211 | 37 (32–42) | 36 (32–41) | 41 |
| $CCl_4$ | 53 (46–63) | 53 (45–62) | 44 |
| $CH_3CCl_3$ | 37 (26–52) | 30 (21–43) | 38 |

[a]All lifetimes calculated using CFC-11 lifetimes of 60 years, with CFC-11 lifetimes based on CFC-12 lifetime of 100 (Laube et al., 2013) or 102 (this study) years.

**Table 6.** Updated mid latitude FRFs based on our 'best estimate' mean ages (taken at 3 years) derived in this study, compared to existing literature values

| Compound | This study (min-max) | Laube et al. (2013) | WMO (2014) |
| :---: | :---: | :---: | :---: |
| CFC-11 | 0.47 (0.43–0.52) | 0.35 (0.32–0.39) | 0.47 |
| CFC-113 | 0.30 (0.27–0.34) | 0.22 (0.20–0.25) | 0.29 |
| CFC-12 | 0.26 (0.23–0.30) | 0.19 (0.16–0.21) | 0.23 |
| HCFC-141b | 0.31 (0.27–0.36) | 0.17 (0.14–0.21) | 0.34 |
| HCFC-142b | 0.13 (0.11–0.15) | 0.05 (0.04–0.06) | 0.17 |
| HCFC-22 | 0.13 (0.11–0.15) | 0.07 (0.05–0.08) | 0.13 |
| Halon-1301 | 0.39 (0.35–0.43) | 0.26 (0.24–0.29) | 0.28 |
| Halon-1211 | 0.66 (0.61–0.71) | 0.52 (0.48–0.56) | 0.62 |
| $CCl_4$ | 0.76 (0.66–0.86) | 0.42 (0.39–0.46) | 0.56 |
| $CH_3CCl_3$ | 0.69 (0.64–0.75) | 0.61 (0.56–0.65) | 0.67 |





**Table 7.** Updated ODPs based on 'best estimate' mean ages (taken at 3 years) derived in this study, compared to existing literature values

| Compound | This study (min-max)[a] | | WMO (2014) | Laube et al. (2013)[*] |
| | ODP | % difference relative to WMO | | |
| --- | --- | --- | --- | --- |
| CFC-11 | 1, by definition | - | 1 | 1 |
| CFC-113 | 0.68 (0.61–0.76) | -20 | 0.81 | 0.63 (0.57–0.69) |
| CFC-12 | 0.70 (0.62–0.79) | -15 | 0.73 | 0.67 (0.59–0.75) |
| HCFC-141b | 0.083 (0.069–0.10) | -18 | 0.102 | 0.063 (0.051–0.076) |
| HCFC-142b | 0.037 (0.031–0.043) | -34 | 0.057 | 0.019 (0.015–0.025) |
| HCFC-22 | 0.028 (0.022–0.035) | -17 | 0.034 | 0.019 (0.015–0.025) |
| Halon-1301 | 19.0 (17.0–22.0) | -25 | 15.20 | 18.7 (17.0–20.3) |
| Halon-1211 | 5.51 (4.89–6.24) | -20 | 6.90 | 5.8 (5.2–6.5) |
| CCl$_4$ | 0.92 (0.80–1.05) | 28 | 0.72 | 0.82 (0.77–0.87) |
| CH$_3$CCl$_3$ | 0.13 (0.11–0.14) | -11 | 0.14 | 0.14 (0.13–0.16) |

[a] min and max values derived from min and max lifetimes and FRF values from Tables 5 and 6. Based on CFC-11 lifetime of 60 years.