# Peer review of "Evaluation of stratospheric age-of-air from CF4, C2F6, C3F8, CHF3, HFC-125, HFC-227ea and SF6; implications for the calculations of halocarbon lifetimes, fractional release factors and ozone depletion potentials."

_Atmospheric Chemistry and Physics, 2017_

## Referee Comment (RC1) · Anonymous Referee #1 · 15 Oct 2017

This paper considers the use of a set of long-lived chemical species that might be alternatives to SF6 and CO2 as 'age-of-air tracers'. The uncertainties involved in the corresponding age-of-air calculations are assessed. A 'best-estimate' mean age calculated from five of the species is compared to that calculated from SF6 and shown to be in good agreement for ages less than about 3 years, but systematically less than that calculated from SF6 for ages longer than that. It is suggested that this difference is likely to be the effect of mesospheric depletion of SF6 and the fact that air samples

taken in the stratosphere inevitably contain a small proportion of air that has previously visited the mesosphere.

Stratospheric lifetimes, fractional release factors and ODPs of a set of ozone-depleting species are calculated on the basis of the 'best-estimate age' and compared to the values from previous similar calculations with SF6-based age of air and to the values recommended in WMO2014.

This paper seems a worthwhile and interesting contribution to me and I recommend publication after revision to address the detailed comments below.

l4-5: 'proxy for the rate of the stratospheric mean meridional circulation' — 'proxy for' and 'rate of' seem odd (second more than first). 'measure' or 'indicator' 'of the strength of' would be more usual.

l21: 'The reduction in SF_6 lifetime' should surely be 'The evidence for reduction in SF_6 lifetime'.

l41: 'it must therefore be reliable ... throughout the stratosphere' — actually by 'reliable' you mean 'largely chemically inert' (the term you have used on l14), so I suggest you use the latter term. Actually 'largely chemically inert' could surely be more precisely stated as something like 'rate of chemical change in stratosphere (and mesosphere) is much smaller than rate of chemical change in troposphere'?

l71: 'We believe the lifetime ...' — you should give at least a very brief indication of WHY you believe this.

l81: 'all compounds' > 'all of the seven compounds to be considered'

l100: 'agrees very well with Advanced Global Atmospheric Gases Experiment' — give a reference for this experiment or the data that comes from it.

Figure 1: The black vertical lines are very small.

l134: To be clear, are you saying that the CF4, C2F6 and SF6 data from AGAGE was

[Figure]

NOT used?

l136: 'in this manuscript' > 'used in this manuscript'

l169: 'Mean ages were calculated using the parametrisation described in Boenisch et al (2009).' — actually Boenisch et al (2009) say 'This two step method that we applied here for stratospheric mean age of air calculation from SF6 observation is explained in detail by Engel et al. (2006b).' — so you should surely give the Engel et al (2006b) as the reference for the method used? But the way in which you provide information on the method used is generally rather confusing and needs to be improved. In the following paragraph you give some further comments on the method and refer to another paper by Engel et al (2002). Then you give further details in section 3c — which to some extent repeats what has already been said in the paragraph l195-202. I think that it is very important to give these sorts of details of the method (including testing the sensitivity to the value of the ratio widthˆ2/mean age). But at present the way that these details is disjointed and, as noted previously, the references to previous work, where the reader might find more detail are not very clear.

l176: 'using values' > 'using values of the above ratio'

l260: I've already noted that this text repeats to some extent what was said in l189-202. It is not necessarily a bad thing to repeat important points, but as noted earlier, I think that the whole presentation of methods could be clearer. Perhaps, for example, it would be more effective to combine the description of each part of the baseline method with the method(s) for the corresponding uncertainty test in Section 2, and then discuss the results of the uncertainty tests and make further comments in Section 3.

l297: 'We use CFC-11 as a vertical coordinate because it is an inherent property of the measured air parcel and will be similarly influenced by transport and mixing' — 'similarly' to what — I guess that you mean 'similarly to the other six tracers' but please clarify. In any case I don't really follow the logic here — aren't the other six tracers also 'inherent properties of the measured air parcel' — so why is CFC-11

special? (I don't see a problem with the use of CFC-11, I just don't follow the logic.'

l393: It would be helpful if you included a brief comment on the information that was used to generate the WMO (2014) recommended values of lifetimes, FRFs and ODPs. Was this a combination of model and observational information? How did it differ from the information used to generate the values in Laube et al (2013)?

———————————————————

---

## Referee Comment (RC2) · Anonymous Referee #3 · 18 Nov 2017

This is an interactive review of the paper titled "Evaluation of stratospheric age-of-air from CF4, C2F6, C3F8, CHF3, HFC-125, HFC-227ea, and SF6; implications for the calculations of halocarbons, fractional release factors and ozone potentials", by Elvidge et al for the journal Atmospheric Chemistry and Physics Discussions. This paper represents an important work that presents the tropospheric trends and stratospheric measurements of 7 trace gases, and their potential to estimate stratospheric mean ages and should be published. Their measurements confirm the results of Ray

et al. (2017) that the atmospheric lifetime of SF6 should be reduced from 3200 years, and it is more like the lifetime of HFC-227ea. Another important result of this paper is that the reported atmospheric lifetime of HFC-125 may be wrong. The error analysis for the age of air, new estimates of FRFs and ODPs, and new stratospheric lifetimes for many ODSs also are of value to the scientific community.

Minor points to address:

1. The authors point out the potential troubles using CO2 as a "mean age of the stratospheric air mass" tracer, because of its strong seasonal cycle and hydrocarbon source. But, there is also a small mesospheric sink for CO2 that produces CO. What is the best literature estimate for the atmospheric lifetime of CO2? Infinite? Cannot these potential effects be easily estimated or considered small? It seems that CO2 is still the best estimate of mean age of air, because it has an infinite atmospheric lifetime. 2. The trace gas, SF6, still is an excellent mean age of air in regions outside the influence of polar air masses and fine for polar air during periods without vertical descent. The qualitative evidence to suggest potential SF6 outside the polar vortex is weak, unless you model the transport. I would recommend dropping it. 3. What are the sinks for these seven gases? Mesospheric sink? Can the Ray et al., (2017) technique be used to calculate their lifetimes too? 4. If the recommended lifetime of HFC-125 is questioned by this work, could the recommended lifetime of HFC-227ea also be wrong. Perhaps the HFCs are not the best lifetime standard after all to compare to SF6. 5. I don't agree with the sentence in the text, how does qualitative evidence go to substantial evidence. I suggest the following "However, we do provide additional new evidence for the need of caution when using SF6 to derive mean ages, particularly in regions influenced by polar vortex descent (Ray et al., (2017)."

---

## Author Comment (AC1) · 18 Dec 2017

**Evaluation of stratospheric age-of-air from CF$_4$, C$_2$F$_6$, C$_3$F$_8$, CHF$_3$, HFC-125, HFC-227ea and SF$_6$; implications for the calculations of halocarbon lifetimes, fractional release factors and ozone depletion potentials**
**ACPD https://doi.org/10.5194/acp-2017-748**
**Response to reviewers**
**Emma Leedham Elvidge et al.**

The authors would like to thank both reviewers for taking the time to review the manuscript, and for their favourable and helpful comments. In particular, the comments that improve the structure of the complex uncertainty analysis performed in this manuscript are welcomed. Our responses to the reviewer comments are given below.

Reviewer comments in blue.
Our responses in black.
A track changes version of the manuscript is available.

**Response to Anonymous Referee #1**

l4-5: 'proxy for the rate of the stratospheric mean meridional circulation' 'proxy for' and 'rate of' seem odd (second more than first). 'measure' or 'indicator' 'of the strength of' would be more usual.
We have changed this sentence to:

important derived quantity used in several stratospheric research fields, often where direct physical or chemical measurements are scarce, not available or inadequate. AoA is perhaps best known for being a  measure of the strength  of the stratospheric mean meridional circulation, the Brewer-Dobson circulation (BDC), as well as being used to determine air mass fluxes between the troposphere and stratosphere (Bönisch et al., 2009). It is also used in

L21: 'The reduction in SF$_6$ lifetime' should surely be 'The evidence for reduction in SF$_6$ lifetime'.
We have changed this sentence to:

research suggests its lifetime has likely been overestimated, thus it may be giving high-biased mean ages. The evidence for a proposed reduction in SF$_6$ lifetime comes from both modelling and measurement studies, which have evaluated its stratospheric loss mechanisms via electron attachment (Kovács et al., 2017) and in the polar vortex (Andrews et al., 2001; Ray et al., 2017). The most recent (at time of writing) evaluation gives a revised lifetime of 850

l41: [1] 'it must therefore be reliable ... throughout the stratosphere' actually by 'reliable' you mean 'largely chemically inert' (the term you have used on l14), so I suggest you use the latter term. [2] Actually 'largely chemically inert' could surely be more precisely stated as something like 'rate of chemical change in stratosphere (and mesosphere) is much smaller than rate of chemical change in troposphere'?
[2] We believe that 'largely chemically inert' (on l14) succinctly sums up our requirements without being overly wordy, and would like to leave this sentence as it is.

[1] We have updated line 41 to reflect the changes suggested above:

the BDC (Mahieu et al., 2014; Stiller et al., 2017). For this reason, i chemical tracer**s are** to be used to diagnose global changes to the BDC they, therefore, must be  chemically inert  throughout the stratosphere. Unfortunately, the influence of SF$_6$-depleted mesospheric air in the upper stratosphere (potential temperature >800 K) and the higher Southern Hemisphere latitudes (poleward of 40 °S) may bias SF$_6$-derived mean ages in these regions (Stiller et al., 2017).

l71: 'We believe the lifetime...' you should give at least a very brief indication of WHY you believe this.

We have updated this section to:

to HFC-227ea than previously thought (Ray et al., 2017, Table 1) and so we include it in our comparison. Finally, we included HFC-125 as a potential age tracer as we believe its current estimated stratospheric lifetime of 351 years (SPARC, 2013, based on model outputs) is potentially an underestimate, based on- preliminary mean age interpretations at UEA (finalised data included later in this manuscript).

l81: 'all compounds' > 'all of the seven compounds to be considered'

We have changed this sentence to:

Anglia (UEA) has analysed whole air samples from the Cape Grim Baseline Air Pollution Station in Tasmania, Australia (https://agage.mit.edu/stations/cape-grim), since 1978, for all compounds discussed in this manuscript except $CF_4$. The Cape Grim (CG) air archive contains trace gas records known to be representative of unpolluted

l100: 'agrees very well with Advanced Global Atmospheric Gases Experiment' give a reference for this experiment or the data that comes from it.

The AGAGE dataset had been referenced earlier, but we realise this might not have been clear for people unfamiliar with the AGAGE set up. Hopefully this re-wording will clarify:

1998), and HFC-227ea (Laube et al. 2010a; Ray et al., 2017). UEA HFC-125 has not been published previously, but the UEA data agrees very well with the CG observations made by AGAGE (Advanced Global Atmospheric Gases Experiment (AGAGE) CG observations, see website link above, (data not shown). Data from high frequency in-situ

Figure 1: The black vertical lines are very small.

This has been improved for the final version of the manuscript.

l134: To be clear, are you saying that the CF4, C2F6 and SF6 data from AGAGE was NOT used?

AGAGE data were used, but just their raw data and not a fit (unlike the UEA data). The sentence has been re-written to clarify this:

fit-derived mean ages was smaller than those derived from the 'raw' CG dataset (S2). As the SIO CG records had a higher sampling frequency during the period of interest only their raw time series – not fitted datasets – were used as inputs into the AoA routine.

l136: 'in this manuscript' > 'used in this manuscript'

Done.

l169: [1] 'Mean ages were calculated using the parametrisation described in (Bönisch et al., 2009)Boenisch et al (2009).' actually Boenisch et al (2009) say 'This two step method that we applied here for stratospheric mean age of air calculation from SF6 observation is explained in detail by Engel et al. (2006b).' so you should surely give the Engel et al (2006b) as the reference for the method used? But the way in which you provide information on the method used is generally rather confusing and needs to be improved. [2] In the following paragraph you give some further comments on the method and refer to another paper by Engel et al (2002). [3] Then you give further details in section 3c which to some
extent repeats what has already been said in the paragraph l195-202. I think that it is very important to give these sorts of details of the method (including testing the sensitivity to the value of the ratio widthˆ2/mean age). But at present the way that these details is disjointed and, as noted previously, the references to previous work, where the reader might find more detail are not very clear.

Organising the discussion of the range of methodological tests in this manuscript was tricky and we agree with the reviewer comments here. We have made several changes to hopefully improve these sections.

To answer [1] and [2] we have re-worded the following part of Section 2:

A sample of stratospheric air represents a mixture of air masses with different transport histories and thus different ages. This distribution of transport times is the 'age spectrum', a probability density function for which the first moment, or mean, is the mean age for that parcel and the second moment, or variance, is the width of the age spectrum (Hall and Plumb, 1994). Mean ages were calculated using the method described in Engel et al. (2002) based on the method provided for inert tracers by Hall and Plumb (1994). This method has been further discussed and modified in various publications, including Engel et al. (2006, 2009), parameterisation described in Bönisch et al. (2009) and Laube et al. (2013). Where we use or refer to the methodological tests or variations used in the papers subsequent to Engel et al. (2002) we will reference these explicitly. To calculate mean age one requires a tropospheric trend, stratospheric measurements and an understanding of the width of the age spectrum. As this study focuses on assessing potential new age tracers we carefully considered the uncertainties associated with the mean ages calculated by our AoA routine. This uncertainty analysis is described in Section 3, where we consider the uncertainties associated with the main inputs to the AoA routine.

To answer point [3] we changed Section 3c to:

**3c. Comparing different methods for implementing the tropospheric time series component of the mean age calculation**
We used an AoA routine based on the algorithm described in Engel et al. (2009), based on the method provided for inert tracers by Hall and Plumb (1994). OneThe limitation of the AoA routine used in this study of this method is that only a quadratic function can be applied for fitting the tropospheric time series for the AoA calculation. A recent improvement is to calculate AoA by a numerical method that uses the convolution of the age spectra, approximated by an inverse Gaussian distribution with the tropospheric time series (Ray et al., 2017), which overcomes the limitations of a quadratic fit to approximate such trends. We implemented this numerical convolution method in our AoA routine

We also checked the rest of the manuscript, namely Sections 3a, b and d, to ensure they provided a consistent message as to the methodology used in this manuscript. The following changes were made to Section 3d:

**3d. Uncertainty in parameterisation of width of age spectrum**

As described in Engel et al. (2002), stratospheric mixing ratios cannot simply be calculated by propagating the tropospheric trend into the stratosphere: due to nonlinearities in the tropospheric trends for our compounds of interest, the width of the age spectrum impacts the propagation of tropospheric trends into the stratosphere. The width of the age spectrum cannot be measured directly and we assume a constant value of 0.7 as the parameterisation of the ratio $\frac{width\ age\ spectrum^2}{mean\ age}$ (from Hall and Plumb, 1994, as used in Engel et al., 2002 and Laube et al., 2013). As described in Sect. 2, we used a value of 0.7 as the parameterisation of the ratio between the squared width of the age spectrum and the mean age to assist with the propagation of non-linear tropospheric trends into the stratosphere. Previous studies have investigated the effect of varying this parameterisation. Engel et al. (2002) investigated the impact of using values of 0, 0.7 and 1.25 and found differences of less than half a year for $CO_2$ and $SF_6$ mean ages. They also reported that the best agreement between these two age tracers was reached when using 0.7. Laube et al. (2010b) also tested the impact of this value on calculated Fractional Release Factors (FRFs, see Sect. 5), comparing values of 0.5, 0.7 and 1.25 and found this factor had a small impact on the FRF for a range of long-lived halocarbons. As this study introduces new potential age tracers, investigating the impact of this parameterisation is pertinent. Values of 0.5 and 1 were compared to the commonly-used value of 0.7 (residual plot in S3). The results are shown in Table 3: one can see that the impact is small (< 1 month, on average) compared to the impact of (a) and (b), and is similar for all compounds.

l176: 'using values' > 'using values of the above ratio'
This sentence was removed as part of the changes addressing the previous point.

l260: I've already noted that this text repeats to some extent what was said in l189-202. It is not necessarily a bad thing to repeat important points, but as noted earlier, I think that the whole presentation of methods could be clearer. Perhaps, for example, it would be more effective to combine the description of each part of the baseline method with the method(s) for the corresponding uncertainty test in Section 2, and then discuss the results of the uncertainty tests and make further comments in Section 3.
This has been answered in our response to the point raised about line l169.

l297: 'We use CFC-11 as a vertical coordinate because it is an inherent property of the measured air parcel and will be similarly influenced by transport and mixing' 'similarly' to what I guess that you mean 'similarly to the other six tracers' but please clarify. In any case I don't really follow the logic here aren't the other six tracers also 'inherent properties of the measured air parcel' so why is CFC-11 special? (I don't see a problem with the use of CFC-11, I just don't follow the logic.
We hope these changes will address this point:

The two key uncertainties from Sect. 3, namely those associated with the tropospheric trend and stratospheric measurements (columns a and b in Table 3), were combined and used as the error bars in Fig. 3, which shows a vertical profile of the mean ages derived from all six of our tracers. We use CFC-11 instead of height or potential temperature as a vertical coordinate because it has a well-quantified vertical distribution because it (Hoffmann et al., 2014) influenced by the same is an inherent property of the measured air parcel and will be similarly influenced by localised transport and mixing processes as our observed age tracers. Tropospheric CFC-11 mixing ratios have slowly declined in the period covered by the stratospheric campaigns (1999-2011) at a rate of between 0.5-1% per year (based

l393: It would be helpful if you included a brief comment on the information that was used to generate the WMO (2014) recommended values of [1] lifetimes, [2] FRFs and [3] ODPs. Was this a combination of model and observational information? How did it differ from the information used to generate the values in Laube et al (2013)

Firstly, we amended the introductory part of Section 5 help address this point:

**5. Implications for policy-relevant parameters**
Younger mean ages do have implications for three important policy-relevant parameters that are used to quantify the impact of halocarbons on stratospheric ozone:
   a.  Stratospheric lifetimes of ODSs.
   b.  FRFs: the fraction of a halocarbon that has been converted into its reactive (ozone-depleting) form in the stratosphere. Compounds with larger FRFs result in greater ozone depletion.
   c.  ODPs: a measure of the impact of individual halocarbons to deplete ozone relative to CFC-11.

In Laube et al. (2013) tThese three parameters were calculated using $SF_6$-based mean ages. Here we updated the Laube et al. results, calculating were calculated using $SF_6$-based mean ages in Laube et al. (2013). We revisit this dataset here, comparing the Laube et al. results to updated FRFs, lifetimes and ODPs calculated usingfrom our new 'best estimate' mean age derived from our five new age tracers for the following 10 ODSs: CFC-11, CFC-113, CFC-12, HCFC-141b, HCFC-142b, HCFC-22, Halon-1301, Halon-1211, carbon tetrachloride ($CCl_4$) and methyl chloroform ($CH_3CCl_3$). We also compare these results to the WMO (2014) recommendations.

Secondly, we addressed [1] by adding the following to the section on stratospheric lifetimes (5a):

recommendations from WMO (2014). In WMO (2014) the stratospheric lifetimes are taken from model-mean values (with the exception of $CCl_4$ where they used tracer and model-mean data) from SPARC (2013). As our lifetime calculation only produces lifetimes relative to that of CFC-11, changes are generally small. The exceptions are the three main hydrochlorofluorocarbons (HCFCs), for which the lifetime has decreased significantly, and $CH_3CCl_3$ for

Thirdly, we addressed [2] by adding the following to the section on FRFs (5b):

Two updates to the FRFs reported in Laube et al. (2013) were made and the resulting FRFs can be seen in Table 6, alongside previous the original Laube et al. results and recommendations from WMO (2014) values based on model-derived FRFs from (Newman et al., 2007). The first change was to use our new 'best estimate' mean age in the FRF calculation. The second change was to use the new methodology outlined in Ostermöller et al. (2017). Based on the work of Plumb et al. (1999) they presented a new formula to calculate FRFs that considers the dependency of the age

Finally, we believe that [3] was already addressed in detail in Section 3c, but hope our changes to the introduction, outlined above, clarify this.

**Response to Anonymous Referee #3 (there is no #2)**

   1.  The authors point out that the potential troubles using CO2 as a "mean age of the stratospheric air mass" tracer, because of its strong seasonal cycle and hydrocarbon source. But, there is also a small mesospheric sink for CO2 that produces CO. What is the best literature estimate for the lifetime of CO2? Infinite? Cannot these potential effects be easily estimated or considered small? It seems that CO2 is still the best estimate of mean age of air, because it has an infinite atmospheric lifetime.

Firstly, we do not provide a detailed comparison or try to discredit $CO_2$ as an age tracer in this paper as we do not have $CO_2$ measurements. $CO_2$ is mentioned in the introduction as it is one of the two age tracers (alongside $SF_6$) people will be most aware of. We felt that an introduction to the topic would not be complete without introducing it. Yes, there is a mesospheric sink for $CO_2$ that produces CO, although this may be considered reversible in the stratosphere where $CO_2$ is reproduced by the reaction of CO with OH radicals (Engel et al., 2006). However, loss processes are not the only factor affecting suitability as an age tracer. When we discuss the fact that no current age tracer is perfect our points regarding $CO_2$ are that one needs to be careful because of its complicated tropospheric trend and its stratospheric source (lines 17-19). This has been stated succinctly in the recent paper by Diallo et al. (2017): *"With the influences of steady growth and seasonal variation, CO₂*

*concentrations in the atmosphere contain both monotonically increasing and periodic signals that represent stringent tests of stratospheric transport and stratosphere–troposphere exchange (STE) in models…"*. We were very clear in our manuscript not to wholly discredit $SF_6$, and we do not discredit $CO_2$ at all. Note line 26: *"These limitations do not preclude the use of $CO_2$ and $SF_6$ as age tracers"*, and later sections (e.g. lines 315-338) discuss only the potential lifetime reduction (already discussed by Ray et al., 2017) of $SF_6$.

Our main aim with respect to the introduction of new age tracers is outlined in the paragraph beginning on line 448: *"The new tracers identified here are not meant to replace $SF_6$ and $CO_2$, which are established age tracers with well-defined tropospheric trends and a wealth of stratospheric measurements, in particular as they are measurable by satellite (Stiller et al., 2008). […] **As future changes to the BDC are likely to be complex, a suite of tracers may be better suited** than $SF_6$ or $CO_2$ **alone** in diagnosing long term changes."*. We believe that the more options we have for potential age tracers the better placed we are. For example, we would like to hope that, one day, the annual increase in atmospheric $CO_2$ may change.

We would also like to draw attention to the 5th paragraph in our introduction that highlights the link between our potential use of 'new' age tracers and the increasing number of methods available for collecting stratospheric air samples, such as AirCores and bag samplers.

We have added a sentence to the above paragraph to stress the long stratospheric lifetime of $CO_2$.

> 2. The trace gas, SF6, still is an excellent mean age of air in regions outside the influence of polar air masses and fine for polar air during periods without vertical descent. The qualitative evidence to suggest potential SF6 outside the polar vortex is weak, unless you model the transport. I would recommend dropping it.

We make no conclusions about this, only saying: *"This raises the question as to whether the sink of $SF_6$ is indeed exclusively located in the mesosphere, although admittedly our non-polar dataset is limited and we cannot rule out mixing of polar vortex air (or vortex remnants) being observed in mid-latitudes outside of the winter polar vortex (Strunk et al., 2000)."*. We would argue that this is a **question**, and a valid one to be raised to prompt future people to model the transport, and would ask to keep it.

> 3. What are the sinks for these seven gases? Mesospheric sink? Can the Ray et al. (2017) technique be used to calculate their lifetimes too?

The PFCs ($CF_4$, $C_2F_6$ and $C_3F_8$ are primarily removed in the mesosphere (above 65 km), mainly by Lyman-$\alpha$ photolysis (WMO, 2014). For HFCs, tropospheric loss via OH is dominant, but losses in the stratosphere come from photolysis and $O(^1D)$ reactions (Naik et al., 2000; Oram et al., 1998; Schmoltner et al., 1993). $SF_6$ lifetimes are discussed in our manuscript lines 21-26 and in Ray et al. (2017). The current, widely-used lifetime of 3200 years is based primarily on loss due to Lyman-$\alpha$ photolysis, but this is now being revised based on our growing understanding of the importance of loss via free electron association in the mesosphere. The method in Ray et al. (2017) – balloon-borne sampler measurements in the polar vortex combined with model outputs – could be used to better quantify mesospheric losses for other compounds that are broken down in this region, if suitable stratospheric datasets exist for these compounds. However, that was not the aim of this study, which uses a mix of polar and non-polar stratospheric data to evaluate potential new age tracers (see our previous responses where we outline the reasons why we believe the addition of new age tracers is important). We hope that the introduction of our combined stratospheric dataset and tropospheric time series, including the uncertainty analysis conducted in Section 3, which highlights the quality of these data, will encourage further exploration of the stratospheric distribution, lifetimes, etc. of these gases.

4. If the recommended lifetime of HFC-125 is questioned by this work, could the recommended lifetime of HFC-227ea also be wrong. Perhaps the HFCs are not the best lifetime standard after all to compare to SF6.

We hope this has partly been answered in our response to reviewer#1. Lifetimes of other HFCs may be incorrect, most are based on model studies (see SPARC, 2013). However, our point was not to correct HFC lifetimes. We have improved our introduction to the HFC-125 lifetime issue in our response to reviewer#1, see above, which explains that preliminary mean age analysis had led us to believe that there may be an underestimation of the HFC-125 lifetime, which we then investigated. As and when further evidence for changing lifetimes of other gases arises we may pursue these avenues as well.

5. I don't agree with the sentence in the text, how does qualitative evidence go to substantial evidence. I suggest the following "However, we do provide additional new evidence for the need of caution when using SF6 to derive mean ages, particularly in regions influenced by polar vortex descent (Ray et al., 2017).

The reviewer did not state which sentence they were referring to, but we assume they referred to line 449 which included the word 'substantial'. The original sentence here: *"However, we do provide substantial new evidence for the need of caution when using $SF_6$ to derive mean ages, especially above the lowermost stratosphere."*

Our results showed that $SF_6$ lifetime does seem to be overestimated, as in Ray et al. (2017). We believe we have substantial evidence to support this, as several new age tracers all show the same result. In the first paragraph of the conclusion the word 'qualitative' refers only to our discussions around **why** $SF_6$ mean ages show a high bias. The reviewer's suggestions of linking our findings to Ray et al. (2017) were already made in the previous paragraph (line 440). We hope the following sentence is a suitable compromise:

The new tracers identified here are not meant to replace $SF_6$ and $CO_2$, which are established age tracers with well-defined tropospheric trends and a wealth of stratospheric measurements, in particular as they are measurable by satellite (Stiller et al., 2008). CO₂, in particular, also has an extremely long stratospheric lifetime. However, the fact that multiple tracers suggest SF₆ mean ages have a high bias suggests we do provide substantial new evidence for athe need foref caution when using $SF_6$ to derive mean ages, especially above the lowermost stratosphere. We also note that, unlike $CO_2$, our new age tracers do not have large seasonal cycles or stratospheric sources and are therefore better suited as tracers of transport times in the lower stratosphere. As future changes to the BDC are likely to be complex, a suite of tracers may be better suited than $SF_6$ or $CO_2$ alone in diagnosing long term changes.

**References:**
The following references were not included in the original manuscript.

Diallo et al. (2017) Global distribution of $CO_2$ in the upper troposphere and stratosphere, ACP, doi:10.5194/acp-17-3861-2017.

Hoffmann et al. (2014) Stratospheric lifetime ratio of CFC-11 and CFC-12 from satellite and model climatologies, ACP, doi: 10.5194/acp-14-12479-2014.

Naik et al. (2000) Consistent sets of atmospheric lifetimes and radiative forcings on climate for CFC replacements: HCFCs and HFCs, JGR, doi:10.1029/1999JD901128.

Oram et al. (1998) Growth of fluoroform ($CHF_3$, HFC-23) in the background atmosphere, Geophys. Res. Lett., doi:10.1029/97gl03483.

Newman et al. (2007) A new formulation of equivalent effective stratospheric chlorine (EESC), ACP, doi:10.5194/acpd-7-3963-2007.

Schmoltner, et al. (1993) Rate coefficients for reactions of several hydrofluorocarbons with hydroxyl and oxygen atom(1D) and their atmospheric lifetimes, J. Phys. Chem., doi:10.1021/j100137a023.